# Probing Interleukin-6 in Stroke Pathology and Neural Stem Cell Transplantation

**DOI:** 10.3390/ijms232415453

**Published:** 2022-12-07

**Authors:** Gavin Miles Lockard, Adam Alayli, Molly Monsour, Jonah Gordon, Samantha Schimmel, Bassel Elsayed, Cesar V. Borlongan

**Affiliations:** 1Morsani College of Medicine, University of South Florida, Tampa, FL 33602, USA; 2Center of Excellence for Aging and Brain Repair, Department of Neurosurgery and Brain Repair, Morsani College of Medicine, University of South Florida, Tampa, FL 33612, USA

**Keywords:** stroke, interleukin-6, stem cells, neuroinflammation, cerebrovascular disease

## Abstract

Stem cell transplantation is historically understood as a powerful preclinical therapeutic following stroke models. Current clinical strategies including clot busting/retrieval are limited by their time windows (tissue plasminogen activator: 3–4 h) and inevitable reperfusion injuries. However, 24+ h post-stroke, stem cells reduce infarction size, improve neurobehavioral performance, and reduce inflammatory agents including interleukins. Typically, interleukin-6 (IL-6) is regarded as proinflammatory, and thus, preclinical studies often discuss it as beneficial for neurological recuperation when stem cells reduce IL-6′s expression. However, some studies have also demonstrated neurological benefit with upregulation of IL-6 or preconditioning of stem cells with IL-6. This review specifically focuses on stem cells and IL-6, and their occasionally disparate, occasionally synergistic roles in the setting of ischemic cerebrovascular insults.

## 1. Introduction to Stroke Epidemiology and Treatment

Globally, stroke is the second leading cause of death, with over 5.5 million people dying from the disease each year [1]. Additionally, stroke represents a major public health burden that costs over 721 billion US dollars annually [2]; over 50% of survivors are significantly disabled, including a loss of cognitive function, linguistic abilities, and motor function [1,3]. Stroke involves disruption of cerebral blood flow (CBF) due to an infarction (ischemic stroke) or vascular rupture (hemorrhagic stroke) [4]. Ischemic stroke is the more prevalent of the two, representing 85% of strokes and carrying a mortality rate of 25%, while hemorrhagic stroke accounts for 15% of strokes and is associated with greater mortality (50%) [3,4]. Stroke can occur in any cerebral blood vessels, with commonly affected arteries being the middle cerebral artery, internal carotid artery, and external carotid artery [5]. Following stroke, the brain undergoes pathological cascades resulting in cell death and neurological dysfunction. Specifically, stroke leads to the activation of immune responses and resulting neuroinflammation, disruption of the blood–brain barrier (BBB), metabolic dysregulation, oxidative stress, and ultimately irreversible neural cell death [3,6]. While the obstruction or hemorrhage characterizing stroke is an acute event, the neurological changes that follow are chronic and contribute to the loss of neurological function [7]. Not all strokes present with a known etiology, but comorbidities and risk factors have been identified, including age, hypertension, obesity, physical inactivity, and diabetes [4].

Despite the prevalence of stroke, there are few treatment options available. Currently, the only FDA approved treatment is tissue plasminogen activator (tPA), a thrombolytic agent used to disrupt blood clots and restore CBF [6]. tPA has clinical utility in breaking up the obstruction causing the ischemia but does little to help the neuropathology that develops secondary to stroke and increases the risk of a hemorrhagic transformation [3,6]. Additionally, only 10% of patients qualify for tPA and it must be administered within approximately 4 h of stroke development [3]. The limited use of tPA underscores the need for novel treatments with a wider treatment window. Thrombectomy, the surgical removal of a clot, is another treatment option that attempts to restore CBF [3]. However, thrombectomy also has a narrow treatment window and is associated with increased morbidity due to its invasive nature [3]. Researchers initially looked towards hyperbaric oxygen therapy (HBOT), which utilizes high pressure and increased oxygen saturation to induce neuroplasticity [8]. The ischemia associated with stroke results in hypoxia, raising the question as to whether additional oxygen can ameliorate damage [6]. While HBOT has shown promise in reducing chronic stroke pathology in some studies, others have posited that it provides no benefit and may even prove harmful [9], highlighting that therapeutics are still lacking.

Recently, there has been interest in targeting the neuroinflammation underlying much of the chronic effects of stroke. Neuroinflammation initially occurs as a protective mechanism to prevent the spread of damage, however, if it is sustained it becomes pathological and exacerbates secondary cell death and excitotoxicity [4]. By targeting neuroinflammation, clinicians and researchers hope to extend the therapeutic window and reduce chronic neurotoxicity and neurodegeneration. One way to achieve this may be stem cells [4], as they have a longer therapeutic window and may simultaneously reduce neuropathology and promote neuroregeneration.

## 2. Interleukin-6 as an Inflammatory Mediator in Stroke

A major mechanism involved in secondary cell death following ischemic stroke is inflammation. In this respect, IL-6 is a unique mediator of this pathophysiology due to its local production in the CNS and its dual signaling pathways to modulate inflammation [10,11] (Figure 1). Soon after ischemic injury, IL-6 promotes an inflammatory environment to remove dead and dying neurons. This initial inflammatory response is beneficial and vital for the eventual induction of IL-6 anti-inflammatory signaling [12,13]. Other members of the IL-6 family include leukemia inhibitory factor (LIF) and ciliary neurotrophic factor (CNF), both of which are anti-inflammatory and regenerative moderators. LIF and CNF prompt neurogenesis, cell survival, and endogenous stem cell proliferation [12,14]. IL-6 has two unique signaling pathways: classical and trans. The trans signaling pathway, while helpful in modulating debris clean up following stroke, can cause a detrimental cyclical inflammatory reaction if employed for too long. Contrarily, using the classical signaling pathway, IL-6 can induce restoration of tissue [15]. This heterogeneity in signaling makes IL-6 a complicated target for therapeutics because it has both toxic inflammatory and restorative roles in stroke pathophysiology. Nonetheless, a careful examination of IL-6, specifically detailing its timing and dosing regimen post-stroke, may reveal the optimal strategy to exert therapeutic effects. A number of studies, however, have exemplified that manipulating levels of IL-6 can have immense impacts on patient prognosis [16,17,18]. Pre-clinically, blocking or amplifying IL-6 has shown similarly mixed results [19,20,21,22,23,24]. The complexity in controlling IL-6 signaling to improve stroke recovery may be due to precise temporal and environmental-based changes influencing IL-6′s preferred signaling. Thus, while pharmaceutical intervention may be too definitive, cell-based therapies offer unique treatment potential due to cells’ abilities to respond to local environments and up- or down-regulate IL-6 signaling.

## 3. Applications and Benefits of Neural Stem Cells

The use of cell-based therapy is a rapidly expanding field. Stem cells can be obtained at various levels of differentiation, can potentially develop into any cell type in the human body, and have shown successful therapeutic outcomes in stroke and other pathologies. This ability to differentiate is referred to as the cell line’s “potency”, with totipotent and pluripotent having the most potential. Further differentiated stem cells are multipotent, such as mesenchymal and hematopoietic cells, and oligopotent cells, such as neural progenitor cells [25]. Neural stem cells (NSCs) can differentiate into a variety of neural cell types including neurons, astrocytes, and oligodendrocytes [26]. In the setting of stroke therapeutics, ethical challenges complicate the use of neural stem cells because they must be obtained from aborted fetuses or difficult-to-reach parts of the brain such as the subventricular zone of the lateral ventricles and the hippocampal dentate gyrus [27,28]. Recent efforts have focused on deriving NSCs from other, more readily accessible parts of the body. One way is the conversion of embryonic stem cells (ESCs) and induced pluripotent stem cells (iPSCs) into NSCs [29]. Human astrocytes have also been reprogrammed and converted into neural stem cells without passing through a pluripotent state such as iPSCs, which minimizes any risk of teratoma formation along with other potential complications [30]. More recently, human fibroblasts were converted into induced NSCs (iNSCs) using Zfp521, a zinc-finger transcription factor [31].

There is currently no approved stem-cell-based therapy for stroke, but research into the topic shows potential. For example, mouse stroke models with a middle cerebral artery occlusion (MCAO) that were treated with reperfusion followed by human NSCs showed significantly reduced infarct volume and improved behavioral outcomes when compared to MCAO models that were treated with reperfusion only [32]. Similarly, NSCs derived from human iPSCs significantly reduced neurological deficits in MCAO mice even when administered 24 h after the occlusion, which offers a much wider therapeutic window than the current standard of care [33]. When tPA was administered to MCAO mouse models outside of its therapeutic window, the subsequent administration of human NSCs reduced the impact of ischemia-reperfusion injury, suggesting synergistic effects with other stroke therapies [34]. Other benefits of NSCs include their ability to reduce ischemia-induced inflammation and induce angiogenesis. For example, MCAO mouse models treated with neural progenitor cells showed a significantly reduced presence of CD45+ cells 30 days post-treatment, and a reduction in MHCII expression at 3 and 10 days post-treatment when compared to untreated mice [35]. Furthermore, rat MCAO models treated with human NSCs displayed higher levels of focal angiogenesis than their control counterparts [36]. All of these findings point to an extremely promising future in NSC therapeutics.

## 4. Mechanisms of Stem Cell Therapy for Stroke

There are multiple mechanisms by which NSCs exert their neuroprotective and therapeutic effects following ischemic stroke. Following stroke, there is an increase in endogenous neural stem cells and associated angiogenesis, aiming to restore cerebral blood flow [37]. Artificially, exogenous administration of NSCs after stroke increases levels of peri-infarct zone angiogenesis [38]. This angiogenesis enhances neurogenesis through factors released by endothelial cells which promote self-renewal and endogenous production of NSCs [39].

The therapeutic effects of NSCs are also mediated by enhanced neurogenesis at the site of injury. Implanted human NSCs in rats induce endogenous NSCs to differentiate into neurons, which is also associated with functional recovery [40]. Direct differentiation of the implanted stem cells is also possible. Intravenously injected human NSCs migrate to the site of injury and differentiate into both neurons and astrocytes [41]. Much of the therapeutic effects of NSCs are mediated by secondary effects such as attenuation of inflammation along with direct differentiation [33].

Another therapeutic potential of NSCs involves mitochondrial recovery. One of the hallmarks of ischemic injury is mitochondrial disfunction, leading to decreased ATP, increased Ca^2+^ in the cell, and the generation of reactive oxygen species. The ability of stem cells to transfer their mitochondria to cells with impaired function has been demonstrated across a wide range of stem cell types and pathologies [42,43]. Transplanted MSCs induce mitochondrial transfer to injured cerebral microvasculature, leading to partial restoration of mitochondrial function and improvements in infarct volume and functional recovery. Several signals are thought to mediate this mitochondrial transfer, including damaged mitochondria, damage-associated molecular patterns (DAMPs) such as mtDNA, ROS and apoptotic markers, and cytokines such as TNFα and NF-κB [42,44,45,46,47].

Transplanted NSCs attenuate inflammation triggered by the activation of astrocytes and microglia [48]. Post-transplantation, genes for inflammatory cytokines IFN-γ, TNF-α, IL-1β, and IL-6 are downregulated, indicating lower levels of inflammation. The injured area also showed lower levels of CD45 30 days post-treatment and lower levels of IBA-1, and IBA-1+/major histocompatibility complex class II 3- and 10 days post treatment. CD45 is a surface marker of microglia that can be used to estimate their presence. Similarly, the level of MHC II is an indicator of an inflammatory response and can also be used to estimate microglia. Thus, the results signify less activated microglia post-treatment. Overall, the attenuation of the immune response decreased glial scarring and secondary injury following stroke [35]. There was also a reduction in the number of GFAP-positive astrocytes, which also contribute to scarring [49]. Transplanted NSCs have also been found to inhibit T-cell proliferation while inducing apoptosis of active T-cells through death signals and other mediators [50,51,52].

While all these aforementioned effects are central and at the site of injury, immunomodulation following stem cell implantation is also regulated peripherally. Within secondary lymphoid organs, dendritic cell antigen presentation and T-cell proliferation are inhibited after NSC transplantation [53,54]. IL-6 phosphorylates an activator and signal transducer of STAT3, which induces the differentiation of T_H_17 cells peripherally. This process is inhibited by LIF, which is released by NSCs after transplantation and functions through a SOCS3 inhibitory pathway [55]. The spleen is also a major site of peripheral inflammatory signaling following ischemic stroke which has been found to attract transplanted neural stem cells [56]. While the role of the spleen and peripheral inflammation is not completely understood, potential therapeutics may be less invasive compared to those targeting the brain [4].

### An Introduction to Interleukin-6 Signaling in Stem Cells

Cell-based therapies may be fundamental to inducing ideal IL-6 signaling in stroke recovery. When treated with human embryonic stem cell-derived and umbilical cord MSCs, animal models of ischemic stroke show decreased IL-6 mRNA relates to increased neurogenesis, angiogenesis, and decreased apoptosis [57,58]. Contrarily, upregulation of IL-6 is important in promoting regulatory T-cell (Treg) differentiation, a subset of anti-inflammatory T-lymphocytes. Bone marrow-derived stem cells (BMSCs) increase T-reg populations, allowing for neuroprotection and anti-inflammation effects in vitro [59]. When oligodendrocyte precursor cells (OPCs) are cultured with BMSCs and Tregs, OPCs show greater myelination potential and increased IL-6 and FGF-β production compared to OPCs cultured with just BMSCs. This study exemplifies OPCs potential to respond to cells in the ischemic environment, supporting their role in modulating the optimal IL-6 signaling pathways [60]. Pre-conditioning studies further demonstrate stem cells immense adaptability and potential to shift IL-6 signaling pathways [61,62,63]. It is apparent that stem cells are able to adapt to the local environment and surroundings in order to increase or decrease IL-6 levels. Interestingly, whether up- or down-regulated, cell-based alterations of IL-6 concentrations showed benefit in the animal models. While concentrations are important to understanding the therapeutic mechanism of stem cells, this research niche would be strengthened by further examining whether IL-6 signaling pathways are altered by stem cells as well, or just the cytokine’s concentration.

## 5. Stem Cells and Interleukin-6: The Occasionally Disparate, Occasionally Cooperative Roles in Ischemic Stroke

A comprehensive review of the pertinent preclinical literature is demonstrated in the text below as well as in Table 1. Common themes are stem cells’ capability to decrease infarction volume, improve neurological behavioral performance, and decrease proinflammatory cytokines such as IL-6, but increased concentrations of IL-6 or IL-6 preconditioning on stem cells has also been shown to increase stem cell performance and neurological outcomes. This largely depends on the timing of transplantation following stroke.

### 5.1. Preclinical Studies Describing the Synergistic Role of IL-6 in the Setting of Stroke

IL-6 may be vital to stroke recovery and stem cell functionality, as exemplified by the following studies. The efficacy of mesenchymal stem cells (MSC) can be tied to IL-6 expression. Gutiérrez-Fernández et al. 2011 examined the impacts of transplantation of mesenchymal stem cells (MSC) in the acute phase of stroke (30 min following occlusion), and sought to identify differences in neurological outcomes depending on the administration route, either venous or arterial. Following the permanent middle cerebral artery occlusion (MCAO) model, rats were administered bone marrow-derived MSCs either intravenously (IV) or intraarterially via the carotid artery. In both delivery methods, the rats demonstrated similar neurological recovery and had decreased cell death compared to control, but stem cell migration and implantation only occurred in the intraarterial route. Neither administration resulted in decreased infarct volume. Vascular endothelial growth factor (VEGF) expression was significantly greater in the arterial administration compared to IV, and compared to control, both administration routes resulted in greater blood vessel numbers 14 days post stroke model. The expression of interleukin-6 was significantly greater in the IV MSC treated rats compared to the IV control rats. Conversely, IL-6 was significantly greater in the intraarterial control rats, compared to the intraarterial MSC rats [64].

In addition to MSCs, IL-6 plays a vital role in NSC efficacy as well. Unfortunately, implanted NSCs death impedes the cells’ therapeutic potential. In an attempt to ameliorate this cell death, IL-6 preconditioning was attempted in order to prompt the STAT-3 survival pathway. Neural stem cells harvested from the subventricular zone of fetal mice were preconditioned with IL-6 and then transplanted in mouse brains following transient MCAO. This preconditioning protected the stem cells from reperfusion injury via STAT3-mediated promotion of manganese superoxide dismutase (mitochondrial antioxidant enzyme). The preconditioning also upregulated secretion of VEGF from the stem cells via STAT3 activation, thus resulting in angiogenesis. The preconditioned stem cells also significantly reduced infarct size and improved neurological performance compared to control stem cells. If stem cells were treated with STAT3 siRNA prior to transplantation, the beneficial effects were lost. Essentially IL-6 preconditioning prepares neural stem cells for the harsh oxidative environment following ischemia-reperfusion [63].

A subset of NSCs, NSC-01, may be superior to MSC. NSC-01 have the unique propensity to extend of filopodia towards ischemic cells. In the setting of oxygen glucose deprivation (OGD) model, primary rat cortical cells and human neural progenitor cells were protected when either was cocultured with NCS-01 cells. Rats were subjected to either transient or permanent MCAO and then received either intracarotid or IV transplants of NCS-01 cells. There were dose-dependent improvements in neurological motor and behavioral function, and reductions in infarct area and surrounding cell loss. The ideal dose and administration was 7.5 × 10^6^ cells/mL via the intracarotid artery within 3 days post-ischemic insult, although there was still benefit when administered within a week. NCS-01 cells were found to secrete basic fibroblast growth factor and IL-6. Interestingly, it was observed that NCS-01 cells extend cadherin-positive filopodia towards ischemic cells [65].

### 5.2. Preclinical Studies Describing the Disparate Role of IL-6 in the Setting of Stroke

Contrary to the beneficial aspects of IL-6 in the studies described above, IL-6 may also amplify inflammation and worsen stroke outcomes. For instance, IL-6 plays a vital role in the post-ischemic microglial recruitment. One day post-MCAO/reperfusion in mice, human neural stem cells were injected into the ipsilesional hippocampus. One day post-transplantation, the neural stem cells had migrated to the lesion, reducing infarct volumes compared to stroke model controls. Behavioral deficits were also improved, and blood–brain barrier damage was ameliorated. Microglia activation was reduced, as were expression of vascular adhesion molecules, including intercellular adhesion molecule-1 and vascular cell adhesion molecule-1. Of particular interest to this study, expression of the following proinflammatory agents were also decreased: tumor necrosis factor (TNF)-α, IL-6, IL-1β, monocyte chemotactic protein-1, macrophage inflammatory protein-1α. Due to the rapid onset of observed benefits, the stem cell effects were felt to be secondary to their anti-inflammatory nature, rather than inherent cell replacement [32]. A similar experiment supports these findings, also demonstrating decreased microglia activation and decreased expression of proinflammatory and adhesion agents [33].

The peripheral inflammatory response following stroke is also heavily reliant on IL-6 signaling, as shown in studies examining spleen involvement. Rats underwent MCAO and 24 h later received either IV saline or IV MAPC derived from human bone marrow. Three days after treatment, RNA was isolated from the lesioned cortex and the spleen. MAPC treatment decreased concentrations of IL-1β and IL-6, and increased IL-10. Regarding the effect of stroke on the spleen, MAPC treatment ameliorated the reduction in spleen mass, as well as increased Treg cells in the spleen. Interestingly, compared to saline control, MAPC treatment enhances neurological recovery in rats with functional spleens, but had no effect in asplenic rats [66].

Additionally, relating to peripheral inflammatory contributions, adipose-derived MSCs on stroke alter IL-6 signaling to restore BBB integrity and reduce peripheral inflammatory cell infiltration into ischemic brain regions. Rats underwent MCAO and then were intravenously administered adipose-derived MSC. Compared to control, this therapy decreased infarct area, cell apoptosis, and expression of proinflammatory IL-1β, IL-6, and TNF-α, and thus decreased blood–brain barrier permeability. Proapoptotic factors in the setting of endoplasmic reticulum stress were reduced, whereas antiapoptotic factors were induced [67]. Matrix-metalloproteinase-9 (MMP-9) is induced in the setting of ischemic cerebrovascular disease, and may also increase blood–brain barrier permeability. Inhibition of MMP-9 by MSC may explain the mechanisms for MSC induced blood–brain barrier recovery. Mice underwent transient MCAO for 90 min and then received intracranial injection of MSC. This therapy decreased infarct volume and improved neurological behavior. It also reduced IgG leakage, tight junction protein loss, and IL-1β, IL-6, and TNF-α. Indeed, it was ultimately seen that MMP-9 activity and protein concentrations were reduced [68].

While it is understood that a permeable BBB may perpetuate ischemic inflammatory damage, one of the standard treatments for stroke may also injure the BBB. Though tPA is a great clot buster, the reperfusion injury following treatment is detrimental and affects the blood–brain barrier; thus, they sought to investigate if human neural stem cells could ameliorate this reperfusion insult following tPA treatment. 6 h following MCAO/reperfusion, mice received IV tPA, and then 24 h after stroke, human neural stem cells were intracranially transplanted. Though tPA is well-utilized treatment in the clinical environment, there is a specific window of 3–4 h, and in this experiment, the delayed tPA administration at 6 h ironically resulted in worsened blood–brain barrier damage and inflammation compared to MCAO controls. However, the neural stem cells were able to attenuate the tPA-induced aggravation of stroke damage. This includes decreased expression of TNF-α, IL-6, and MMP-9, and increased expression of BDNF [34]. Concomitant use of tPA and stem cells may be warranted as a future clinical treatment.

The anti-inflammatory properties of stem cells may be enhanced by preconditioning. Human umbilical mesenchymal stem cells (HUMSCs) primed with VX-765 (an inhibitor of caspase 1) improved neurological outcomes following stroke. Following distal MCAO, mice were transplanted with HUMSCs, or HUMSCs treated with VX-765. Compared to HUMSCs, VX-765-treated HUMSCs had decreased proinflammatory cytokines (including IL-6), apoptosis, and infarction area, and increased anti-inflammatory cytokines and autophagy-related proteins. In vitro, an OGD model demonstrated similar findings with additional findings of decreased p-mTOR and increased p-AMPK, suggesting these pathways have opposing roles in this therapy [58]. AMPK is an enzyme involved in ameliorating neuroinflammation [69]. Furthermore, pretreatment of bone marrow stromal cells with Roxadustat (FG-4592) would promote grafted cell survival. Roxadustat was chosen as this inhibits hypoxia-inducible factor prolyl hydroxylase, thus stabilizing concentrations and activating the pathway of hypoxia-inducible factor-1α, which was previously shown to be beneficial in attenuating rat neuronal apoptosis following ischemic stroke [70,71]. In vitro OGD model demonstrated that Roxadustat pretreatment promoted autophagy to inhibit apoptosis through HIF-1α/BNIP3 signaling pathway. Indeed, Roxadustat ensured greater survival rate of the bone marrow stromal cells, and decreased infarct volume while simultaneously improving neurological behavioral scores. There was also improved surrounding neuronal survival and reduced microglia activation, as well as decreased proinflammatory IL-1β, IL-6, and TNF-α via the TLR-4/NF-κB signaling pathway [61]. Pre-conditioning of stem cells for stroke therapies may significantly amplify their therapeutic potentials.

After review of these studies, it is apparent that IL-6 has differing roles in modulating neuroinflammation following stroke. While the inflammatory nature of IL-6 is beneficial to recovery in some instances, it can be pathological if prolonged. IL-6 signaling can take the classical, homeostatic pathway or the trans, pro-inflammatory pathway. Which version of signaling is employed is influenced by the local post-ischemic environment. If stem cells’ use of IL-6 signaling can be modulated by local signaling as well, stem cells can be employed to amplify whichever signaling pathway would be most beneficial given the time line of the ischemic insult. Further research is needed to best understand which IL-6 signaling pathway is employed at different time points following stroke to induce recovery.

**Table 1 ijms-23-15453-t001:** In vivo Stem Cell Studies Affecting IL-6 Expression. This table outlines cell-based preclinical trials finding regulatory effects on IL-6 expression as a mechanism to enhance ischemic stroke recovery.

Citation	Sample	Time to Transplantation (Post Induction)	Cell Type	Cell Pretreatment	Route	Dosage	Results
Gutiérrez-Fernández et al. (2011) [64]	MCAO rats	30 min	Bone marrow-derived MSCs	-	Intravenous or intra-arterial	2 × 10^6^ cells	Intravenous injection produces the highest levels of IL-6. Both routes decrease apoptosis, enhance angiogenesis, and improve neurological function.
Sakata et al. (2012) [63]	MCAO mice	6 h or 7 days	NSCs	IL-6	Intracerebral	1 × 10^5^ cells	IL-6 preconditioning increases NSC tolerance for oxidative stress and induces angiogenesis.
Huang et al. (2014) [32]	MCAO mice	24 h	hNSCs	-	Intracerebral	1 × 10^6^ cells	Within 24 h of hippocampal transplantation, hNSCs decreased expression of inflammatory cytokines (IL-6, IL-1β and TNF-α, monocyte chemotactic protein-1, macrophage inflammatory protein-1α) and adhesion molecules, reduced BBB damage, and improved behavioral function.
Eckert et al. (2015) [33]	MCAO rats	24 h	hiPSC-NSCs	-	Intracerebral	1 × 10^6^ cells	Downregulation of IL-6, TNF-α, and IL-1β was associated with reduced microglial activation, enhanced BBB restoration, and preserved neurological function.
Yang et al. (2017) [66]	MCAO rats	24 h	Multipotent Adult Progenitor Cells	-	Intravenous	1.2 × 10^7^ cells	MAPC transplant reduced IL-6 and IL-1β while increasing IL-10 in serum. Stroke recovery was dependent on intact spleen, acting by restoring spleen mass reduction caused by stroke.
Chi et al. (2018) [67]	MCAO rats	0, 12, or 24 h	Adipose-derived MSCs	-	Intravenous	2 × 10^6^ cells	BBB permeability decreases by downregulating expression of IL-6, IL-1β and TNF-α. ER stress response is reduced, inducing anti-apoptotic factors. Infarct area and neurological function is ameliorated.
Cheng et al. (2018) [68]	MCAO rats	15 min	Bone marrow-derived MSCs	-	Intracerebral	1 × 10^5^ cells	MSC transplant reduced inflammatory cytokines (IL-6, IL-1β and TNF-α), tight junction protein loss, IgG leakage, and matrix metalloproteinase expression to attenuate infarct volume and neurological function.
Gong et al. (2019) [72]	CA rats	1 h	Adipose-derived MSCs	-	Intravenous	5 × 10^6^ cells	Stimulation of IL-6 and BDNF expression results in improved neurological deficits reduces hippocampal apoptosis.
Kaneko et al. (2020) [65]	MCAO rats	3 h to 1 week	Bone marrow-derived NCS-01 cells	-	Intra-arterial or intravenous	7.5 × 10^6^ cells	NCS-01 cells secrete IL-6 and bFGF while contributing to filopodia formation. This results in improved motor function, neurological behavior, and reduction of infarct volume.
Salehi et al. (2020) [73]	MCAO rats	Immediate	Epidermal neural crest stem cells or bone marrow-derived MSCs	-	Intra-arterial or intravenous	2 × 10^6^ cells	Neural crest stem cell transplantation results in greater expression of IL-6, BDNF, nestin, Sox10, doublecortin, β-III tubulin and GFAP while decreasing expression of nerotrophin-3 and IL-10. Both cell types decreased infarct volume and improved functional outcome.
Boese et al. (2020) [34]	MCAO mice	24 h	hNSCs	-	Intravenous	1 × 10^6^ cells	In delayed tPA treatment, hNSCs decreased expression of proinflammatory factors (including IL-6), decreases matrix metalloprotease expression, increased BDNF expression, and attenuated BBB damage.
Sun et al. (2020) [58]	MCAO mice	1 h	HUMSCs	VX-765	Intracerebral	1 × 10^5^ cells	VX-765 pretreatment downregulates pro-inflammatory cytokines, including IL-6), while upregulating anti-inflammatory cytokines to reduce apoptosis and neuroinflammation.
Chen et al. (2022) [61]	MCAO rats	24 h	Bone marrow-derived MSCs	Roxadustat	Intracerebral	5 × 10^5^ cells	Roxadustat pretreatment decreases IL-6, IL-1β and TNF-α expression, resulting in improved neurological function.
Ranjbaran et al. (2022) [74]	2VO rats	2 h	Adipose-derived MSCs	-	Intraperitoneal	1 × 10^6^ cells	Inhibition of IL-6 and TNF-α results in decreased apoptosis in the hippocampus and improved memory deficits.

## 6. Conclusions

It is well documented in the literature that stem cell transplantation improves neurological recuperation following stroke models in preclinical studies. There is promising benefit seeing that the stem cells may be administered up to 24 h following stroke, compared to tissue plasminogen activator (tPA) having a window of <3–4 h after stroke onset. Additionally, a lot of the deleterious effects of stroke are secondary to the reperfusion injury, which stem cells may uniquely treat where tPA and thrombectomies cannot. Perhaps future therapies will include concomitant tPA and stem cell transplantation. Of particular interest to this review are the effects stem cells and interleukin-6 have on each other, occasionally differing depending on the preclinical trial’s timing of transplantation. IL-6 is an ironic cytokine; typically regarded as proinflammatory but also prosurvival at times, and this discrepancy is attributed largely to timing.

## Figures and Tables

**Figure 1 ijms-23-15453-f001:**
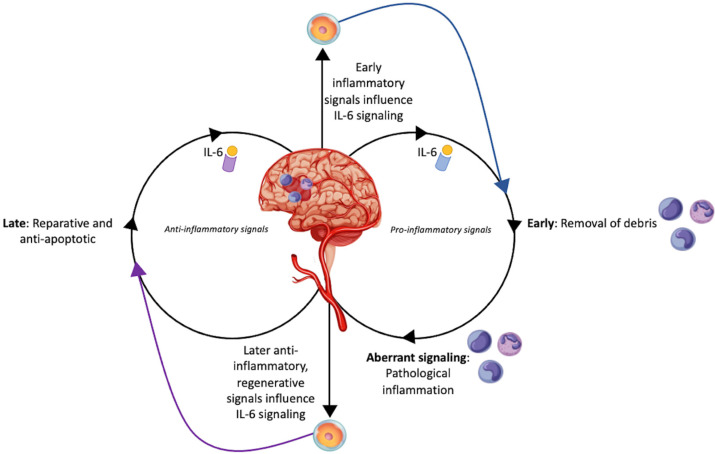
Interleukin-6 modulation by stem cells in stroke. This figure illustrates the dual role of interleukin-6 signaling in stroke pathophysiology. Early signaling of IL-6 can be beneficial in instigating debris clean up after stroke, but prolonged pro-inflammatory IL-6 signaling can be detrimental. Contrarily, late, reparative IL-6 signaling can aid in restoring neurological function. Signaling from the local injured brain tissue may influence stem cells to enhance pro- or anti-inflammatory signaling depending on whether it is early or late in the pathophysiological processes. The diverse potential of stem cells in modulating this signaling is demonstrated as well.

## Data Availability

Not applicable.

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
