# Peer review of "Probing Interleukin-6 in Stroke Pathology and Neural Stem Cell Transplantation"

_ijms, 2022, doi:10.3390/ijms232415453_

Round 1
Reviewer 1 Report
This review provides a brief summary on the therapeutic potential of transplanting neural stem cells and other adult tissue stem cells for the treatment of stroke in pre-clinical animal models through pre-conditioning with IL-6 or alteration of IL-6 expression. In the summary table, the timing of transplantation and whether stem cells are pre-conditioned with IL-6 should be included. The authors should comment why administration of hiPSCs-NSCs led to downregulation of IL-6 expression while bone marrow-derived NSC-01 cells secreted IL-6. Both treatments resulted in the same outcome such as improving neurological function/behavior.
In rows 95/96, Therapeutic use of stem cells in stroke and other pathology is a rapidly expanding field with full of promise.
In row 96, the sentence “Stem cells can be obtained at different levels of differentiation” does not make any sense to me.
What do you mean …an upregulation of NSCs in the infarct region,..”? The term ‘upregulation’ is often used to describe increased level of gene expression instead of cell type.
Explain why reduced presence of CD45+ cells 30 days post-treatment and a reduction in MHCII expression at 3 and 10 days post-treatment are beneficial in MCAO mouse models.
If there is an increase in endogenous neural stem cells following stroke, why is it necessary to administer exogenous NSCs? Why stroke could lead to an increase in endogenous neural stem cells?
The authors mentioned “The majority of the therapeutic effects of NSCs, however, are mediated by secondary effects such as attenuation of inflammation rather than direct differentiation, which is evident through the low expression of neuronal markers in transplanted cells”. I don’t agree with that as there are also numerous studies showing the ability of transplanted NSCs to directly differentiate into neurons at the spinal cord injury site.
Rephrase the following sentence:
This is also contributed to by the reduction of the number of GFAP-positive astrocytes, which also contribute to scarring.
Both paragraphs from rows 238 to 256 show similar content. Combine them together into one paragraph.
Clarify the source of IV human multipotent adult progenitor cells
What is Roxadustat (FG-4592)?
The treatment enhanced Klotho-a and AMPK-a. What are their role in the treatment?
Reviewer 2 Report
Summary
The authors present a comprehensive review of recent research on stem cell transplantation for stroke treatment and the role of IL-6 in this process. The area of research is an important one, thus a review like this is warranted. Sections 1 to 4 are well written, but section 5 needs significant improvement, as commented below.
Comments
1. In section 5 (stem cells and IL-6), the way of presenting previous research is pretty dry and needs to be improved.
1.1. The authors describe each study in a separate paragraph, and start each paragraph with a specific experimental treatment to cells or animals in that study. The way of writing is as if all these studies were part of a single story following some kind of timeline (or a large study conducted by the same people). But these are in fact separate and independent studies. I'd encourage the authors to start each study with something like "In the study by XXX et al, NSCs/mices were treated in certain ways ..." or "XXX et al performed experiments ... ", or any other way to clarify that each paragraph is an independent study conducted by separate research groups.
1.2. There are more than 10 studies described, one after another, but without much logical connection between them. I'd encourage the authors to better group/arrange the studies by area and the questions they try to answer, and use proper transitions between paragraphs.
1.3 In all these paragraphs, it's not obvious why each study was done. The authors merely stated the experimental treatment and the results, but did not mention why a given study was performed (are they all trying to answer the exact same question? What's the unique motivation for each study? Does one study answer something unaddressed in another one?). I'd suggest starting each paragraph with the motivation and unique questions the study tries to address, something like "To understand/address ..., XXX et al did experiment YYY and found ZZZ".
2. After reviewing the studies in section 5, the author made a few simple conclusions as the end of the article. I'd expect the authors to at least (1) present a holistic picture from summarizing these studies (e.g. a mechanistic diagram, Figure 1 was too simple), and (2) draw some new insights or raise high priority questions that motivates further research.
3. Figure 1 is not very intuitive and contains redundant components such as "local signaling influences IL-6 production"
4. Line 79-83 not very easy to read. Please rephrase it.
Round 2
Reviewer 1 Report
The authors have addressed all my concerns.
Reviewer 2 Report
The authors have sufficiently addressed my previous comments.
A minor comment: in line 332, it's not clear what "they" refers to. It's better to be more clear.